# Oils’ Impact on Comprehensive Fatty Acid Analysis and Their Metabolites in Rats

**DOI:** 10.3390/nu12051232

**Published:** 2020-04-27

**Authors:** Agnieszka Stawarska, Małgorzata Jelińska, Julia Czaja, Ewelina Pacześniak, Barbara Bobrowska-Korczak

**Affiliations:** Department of Bromatology, Faculty of Pharmacy, Medical University of Warsaw, Banacha 1, 02-097 Warsaw, Poland; agnieszka.stawarska@wum.edu.pl (A.S.); malgorzata.jelinska@wum.edu.pl (M.J.); julia.czaja.prv@gmail.com (J.C.); ewepaczesniak@o2.pl (E.P.)

**Keywords:** fatty acid composition, vegetable oils, desaturases, arachidonic and linoleic acid metabolites, hydroxyeicosatetraenoic acids, hydroxyoctadecadienoic acids

## Abstract

Fatty acids, especially polyunsaturated, and their metabolites (eicosanoids) play many pivotal roles in human body, influencing various physiological and pathological processes. The aim of the study was to evaluate the effect of supplementation with edible oils diverse in terms of fatty acid composition on fatty acid contents, activities of converting their enzymes, and on lipoxygenase metabolites of arachidonic and linoleic acids (eicosanoids) in rat serum. Female Sprague-Dawley rats divided into seven groups were used in the study. Animals from six groups were fed one of oils daily (carotino oil, made up by combining of red palm oil and canola oil, linseed oil, olive oil, rice oil, sesame oil, or sunflower oil). One group received a standard diet only. Fatty acids were determined using gas chromatography with flame ionization detection. Eicosanoids—hydroxyeicosatetraenoic (HETE) and hydroxyoctadecadienoic acids (HODE) were extracted using a solid-phase extraction method and analyzed with HPLC. Vegetable oils given daily to rats caused significant changes in serum fatty acid profile and eicosanoid concentrations. Significant differences were also found in desaturases’ activity, with the linseed and olive oil supplemented groups characterized by the highest D6D and D5D activity. These findings may play a significant role in various pathological states.

## 1. Introduction

Fats belong to basic nutrients for human body, constituting the main source of energy. A profile of fatty acids is one of the factors affecting their health value. Besides the fatty acid composition, the ratios between saturated, monounsaturated, and polyunsaturated fatty acids (SFA, MUFA, and PUFA, respectively) play an important role. Fatty acids, particularly PUFA belonging to n-6 and n-3 families, are thought to participate in regulation of many physiological and pathological processes, such as inflammation, glycemic control, lipid metabolism, oxidative stress, cardiovascular diseases (CVD), skin changes, asthma, nervous system disturbances, or cancer [1,2,3]. Thanks to research on the biological properties of individual fatty acids, especially PUFA, new mechanisms of their action are discovered. PUFA are believed to exert their effects directly or indirectly through various metabolites [4,5,6]. Thus, they may be used to optimize a diet and prevent a variety of diseases.

Fatty acid contents in the body can be influenced by many factors such as everyday diet, the activity of PUFA converting enzymes (desaturases, elongases) and a general state of an organism. It has been observed that dietary fatty acids, including those originating from various vegetable oils used both in cooking and as dietary supplements, modify the body’s fatty acid profile [7]. Numerous studies indicated the inverse correlation between increased fat intake, especially SFA, and the risk of coronary events [8,9,10]. The replacement of some amounts of SFA with MUFA, administered as olive oil (up to 75% of oleic acid, OL, C18:1, *n*-9), resulted in a significant risk reduction of cardiovascular mortality, various cardiovascular events, and stroke [11]. Similarly, high-oleic acid soybean oil (over 70% of oleic acid vs. 28% in the standard soybean oil), oils rich in n-6 PUFA such as linoleic acid (LA, C18:2, n-6) or in n-3 PUFA like α-linolenic acid (ALA, C18:3, n-3) and its metabolites— eicosapentaenoic acid (EPA, C20:5, n-3), docosahexaenoic acid (DHA, C22:6, n-3)—used to substitute SFA showed significant decrease of total cholesterol (TC), LDL cholesterol, plasma triglycerides, and apolipoprotein B (apoB) as well as of a general reduction of CVD risk [12,13]. N-3 PUFA supplementation or moderate consumption of fish may also decrease the risk of Alzheimer’s disease and dementia [14,15]. Other studies indicate that the intake of n-3 PUFA is associated with a lower risk the head and neck, esophagus, colorectal, and breast cancer [16,17,18,19]. Studies concern fatty acid profile, their metabolites, and the activity of enzymes involved in their conversion.

PUFA metabolites synthesized on cyclooxygenase (COX) or lipoxygenase (LOX) pathways can have a significant effect on many processes in the body. They are referred to as eicosanoids. They are locally active compounds derived from arachidonic acid (AA, C20:4, *n*-6), dihomo-γ-linolenoic acid (DGLA, C20:3, *n*-6), and EPA. The group also includes octadecanoids and docosanoids, which are metabolites of linoleic acid and DHA, respectively [20,21]. Whereas arachidonic acid metabolites synthesized on the COX pathway, such as prostaglandins, belong to well-known compounds, hydroxyeicosatetraenoic acids (HETE) produced on the LOX pathway were initially considered biologically inactive [22]. They are currently regarded as important lipid mediators influencing various pathological processes, such as inflammation, asthma, allergy, diabetes, hypertension, or cancer [23,24]. Depending on the LOX isoform engaged in AA metabolism, three main isomers of HETE exist: 5-, 12-, and 15-hydroxyeicosatetraenoic acids (5-, 12-, and 15-HETE). 12- and 5-HETE affect growth and development of different types of cancer [23]. 12-HETE has been indicated to induce cancer cell proliferation, motility, invasiveness, and angiogenesis, as well as to enhance their adhesion leading to facilitated metastasis [23,25]. It also inhibited apoptosis. 5-HETE, alike 12-HETE isomer, was also noticed to stimulate proliferation and enhance growth of breast, prostate, and lung cancer, acting as a survival factor for tumor cells [24,26]. 15-HETE, another arachidonic acid derivative, seems to be acting in an opposite way, enhancing apoptosis, and inhibiting proliferation of prostate and colorectal cancer cells. Its lower concentration in lung tumors correlated with decreased activity of peroxisome proliferator-activated receptor gamma (PPARγ) and, as a result, with cancer development [27].

12-HETE appears also to play role in human hypertension, atherosclerosis, and diabetes. Its significantly elevated level was observed in hypertension patients, comparing to control subjects [28]. What is more, the urinary secretion of this isoform was enhanced in patients than in the control group. Similarly, increased plasma levels of 12-HETE were found both in diabetic patients and diabetic patients with coronary artery disease, when compared to the control group [29]. HETE may also play an important role in the regulation of allergic reactions. Significantly higher sputum concentrations of 15-HETE were noted in asthma patients, then in healthy persons [30].

In the light of above information, the aim of this study was to estimate the influence of dietary supplementation with six selected oils (carotino oil, linseed oil, olive oil, rice oil, sesame oil, sunflower oil), characterized by different composition and content of fatty acids, on fatty acid profile in serum of rats. Indices of fatty acid desaturase activities were assessed as the product/substrate ratio: Δ6-desaturase (D6D) [18:3 *n*-6/18:2 *n*-6], and Δ5-desaturase (D5D) [20:4 *n*-6/20:3 *n*-6].

The contents of lipoxygenase metabolites of arachidonic acid (5-, 12- 15-HETE), linoleic acid (HODE) and EPA (5-, 12-, 15-HEPE) in rat serum were also determined. Knowledge of the relationship between them and different components of the diet (e.g., edible oils), makes it possible to modify their biosynthesis and thus support the treatment of some diseases.

## 2. Materials and Methods

The investigations were approved by the second Local Ethics Commission at the Medical University of Warsaw dealing with experiments with laboratory animals. Female Sprague-Dawley rats (n = 63, age 30 days), obtained from the Animal Laboratory, Department of General and Experimental Pathology, Medical University of Warsaw, were used in the study.

### 2.1. Animals and Study Design

During the experiment all animals had constant access to water and food (Labofeed H fodder, produced by: Wytwórnia Pasz—A. Morawski, Kcynia) and were kept in a room of constant humidity and temperature (23 °C), where a 12-h light-dark cycle was maintained. Detailed composition of Labofeed H fodder is given in Table 1.

After a one-week adaptation, the rats were randomly assigned to one of seven experimental groups (9 animals in each group). The first group (the control group) was fed the standard rodent diet and 0.4 mL/day of water via gavage. From the 40th to 75th days of life (35 days) the animals in the other 6 groups (n = 54) were given 0.4 mL/day of one of six selected, commercially available vegetable oils (carotino oil, made up by combining of red palm oil and canola oil, linseed oil, olive oil, rice oil, sesame oil, and sunflower oil) via gavage in addition to the standard diet. They were stored at 8 °C before being given to animals. The fatty acid composition of oils used in the experiment is shown in Table 2. Every week the rats’ body weight was assessed, and the animals were examined by palpation. The rats were decapitated after 24 h of starvation and the last oil supplementation in the 7th week of the experiment. Blood and organs were collected and the weight of internal organs was determined. The biomaterial samples were stored at a temperature of −70 °C until the test time.

### 2.2. Preparation of Serum

Serum was obtained by the centrifugation of fresh blood for 10 min at 3000 rpm at 4 °C and stored at −60 °C in Eppendorf test tubes until further analysis.

### 2.3. Fatty Acids Analysis in Serum

Fatty acid analysis was performed with gas chromatography (GC, Shimadzu GC-17A chromatograph equipped with flame-ionization detector). Samples of rat serum (100 μL) were transesterified according to the procedure of Bondia-Pons et al. with slight modifications [31]. Without prior lipid extraction, the samples were hydrolyzed by heating with 2.5 mL sodium methoxide in methanol (0.5 mol/L) at 80 °C and fatty acids were converted to methyl esters by heating with 2.5 mL of 14% boron trifluoride-methanol reagent at 80 °C for 3 min. Fatty acids methyl esters (FAME) were isolated with hexane (2 × 0.5 mL) after adding 1.0 mL of saturated sodium chloride solution. Organic extracts were dried with anhydrous sodium sulphate and evaporated to dryness under a stream of nitrogen. FAME were diluted in 20 μL of hexane and stored at −20 °C. The injector of the gas chromatographer was heated to 250 °C and the detector to 270 °C. Separations of FAME were performer on Rtx-2330 capillary column (40 m × 0.18 mm i.d., film thickness: 0.1 μm, Restek) with helium as the carrier gas. The initial oven temperature was 140 °C for 1 min, thereafter increased by 20 °C/min to 170 °C and held for 20 min and increased by 10 °C/min. to 230 °C held for 15 min. The helium flow through the column was 2 mL/min. The whole analysis lasted 41 min. FAME standards (Supelco 37 Component FAME Mix). Results were expressed as μg of each fatty acid per ml of serum and as percentage of total fatty acids present in serum.

### 2.4. Estimation of Desaturases Activity

The product to precursor FA ratios were calculated to estimate the activities of desaturases [32,33,34]. The index of Δ6-desaturase (D6D) activity was determined as the ratio of γ-linolenic acid concentration (GLA, C18:3, *n*-6) to the linoleic acid (LA, C18:2, *n*-6) concentration in serum. The index of Δ5-desaturase (D5D) was determined as the ratio of arachidonic acid concentration (AA, C20:4, *n*-6) to the dihomo-γ-linolenic acid concentration (DGLA, C20:3, *n*-6).

### 2.5. Lipoxygenase Metabolite Analysis

Solid phase extraction method (SPE) was used to extract HETE, HODE, and HEPE from rat serum, according to the procedure, described previously by Frohberg with slight modifications [35,36]. Methanol (0.5 mL) was added to samples of serum (0.4 mL) which were then diluted with water to achieve 10% methanol solution. The samples were loaded onto the SPE C18 cartridges (Backerbond C18, 500 mg/3 mL, J.T. Baker, Holland), preconditioned with methanol, and followed by water (10 mL each). Cartridges containing applied samples were washed with 2 mL of water and 2 mL of 10% methanol, then compounds to be analyzed were eluted with 100% methanol (3 × 0.5 mL). The samples were evaporated to dryness under the nitrogen stream and redissolved in 100 μL of ethanol.

Eicosanoids were determined with high performance liquid chromatography (HPLC), using Shimadzu system comprising LC-20AD pump, DGU-20A5 degasser, UV–VIS SPD-10AV detector and CTO-10 AS VP oven. The compounds studied were separated on Nucleosil C18 column (Nucleosil C18, 100-5, 250 × 4.6 mm, 5 µm, Macherey-Nagel) held at 30 °C. Samples injected to the HPLC system were eluted with a mobile phase composed of methanol, water and acetic acid (73:27:0.01, by volume), with a flow rate of 0.8 mL/min. Detection wavelength was 235 nm. The whole analysis lasted 45 min.

### 2.6. Statistical Analysis

Statistica 13.0 (StatSoft, Kraków, Poland) was used to conduct a statistical analysis of the results. The normality assumptions were estimated with Shapiro–Wilk’s test and whenever the normality and variance homogeneity assumptions were fulfilled one-way analysis of variance (ANOVA) with post hoc Tukey’s test was used. If the assumptions of the analysis of variance were not met, the non-parametric Kruskal–Wallis test, which is a non-parametric equivalent of one-way ANOVA with post hoc multiple comparison test was used. Spearman’s coefficients were used to estimate potential correlations between serum contents of PUFA and their metabolite concentrations. The accepted significance level was established at *p* < 0.05.

## 3. Results

In the current experiment, we studied the influence of diverse edible oils on fatty acid profile and metabolites of arachidonic acid, linoleic acid and EPA in rat serum, as well as on activity of ∆6- and Δ5-desaturases. The general effect of oils on the rat growth and weights of organs was also observed. The oils used for dietary supplementation differed considerably in terms of the percentage composition of fatty acids (Table 2). In the case of olive oil (O) and carotino oil (CAR) oleic acid was a dominant fatty acid. Its amounts in these oils were as follows: 74.46% and 52.94% of all fatty acids, respectively, whereas linoleic acid was present in the highest content in sunflower oil (SUN), 59.91% of all fatty acids. Linseed oil (L) is a good source of γ-linolenic acid, containing even 63% of this fatty acid. Among all saturated fatty acids, palmitic acid occurred in the greatest quantities in all oils. In rice oil (R), it constituted 19.88% of all fatty acids. This oil also contains moderate amounts of oleic acid (40.90%) and linoleic acid (29.75%). Sesame oil (SES) has got a very similar profile of fatty acids to the rice oil. It consists mainly of 10% of palmitic acid, 36.27% of oleic acid, and 44.21% of linoleic acid.

### 3.1. Fatty Acid Analysis in Serum

Arachidonic, palmitic, linoleic, stearic, and oleic acids occurred in dominating proportions among fatty acids present in serum (Table 3, Figure 1). The highest contents of both stearic acid (21.27 ± 0.60%) and arachidonic acid (33.42 ± 1.25%) were observed in the R group. On the other hand, in this group the lowest levels of palmitic (13.61 ± 0.40%) and oleic acids (6.60 ± 0.60%) were found. The highest level of oleic acid was determined in the CAR group (12.70 ± 3.35%). It was significantly higher than in all other study groups. The group receiving the diet supplemented with olive oil stands out from the others with the lowest content of linoleic acid (11.26 ± 0.70%). Arachidonic acid had the highest percentage among the determined fatty acids in all examined groups. Its highest share was found in the group receiving the diet supplemented with rice oil (33.42 ± 1.25%), while the lowest— control group (C)—was fed the standard diet only (29.12 ± 2.79%) (Table 3, Figure 1).

Dietary supplementation with oils caused a decrease in the content of saturated acids in serum, while in the case of monounsaturated and polyunsaturated acids the change in their amount depended on the type of supplementation. The highest average content of all saturated acids occurred in the C group, while it was slightly lower in the L group, and the lowest—in the CAR group (Table 3).

MUFA occurred in the smallest amount in the total fatty acid pool (ranging from 7.92 ± 0.67% in the R group to 13.42 ± 5.67% in the CAR group) in the all studied groups. On the contrary, the share of PUFA was the largest (from 48.23 ± 1.45% in the O group to 53.27 ± 1.11% in the group R).

The share of monounsaturated acids varies significantly between groups supplemented with rice and carotino oil. A similar relationship occurred in the case of total PUFA (tPUFA) and PUFA from the *n*-6 family (*n*-6 PUFA) in the CAR (49.41 ± 1.27% and 45.07 ± 1.29%, respectively) and R groups (53.27 ± 1.11% and 49.82 ± 0.53%), as well as between the R and O groups. Regardless of the supplementation used, the level of *n*-3 family acids remained at a similar level in all study groups.

### 3.2. Estimation of Desaturase Activity

Activities of Δ6- and Δ5-desaturases in serum of experimental rats were affected by the dietary supplementation (Figure 2A,B). Differences in the desaturase activity were found among experimental groups. The Δ6-desaturase activity tends to be significantly lower in control group compared to groups supplemented with oils. The group that received linseed oil was characterized by the highest D6D activity (0.0394 ± 0.05) (Figure 2A). High D6D activity and the highest D5D activity were found in serum of the rats supplemented with olive oil (Figure 2A,B).

### 3.3. Lipoxygenase Metabolite Analysis

The contents of five hydroxy fatty acids in serum of rats fed with a vegetable oil (olive oil, linseed oil, sunflower oil, rice oil, carotino and sesame oil) were analyzed in the present study. 5-, 12-, 15-HETE are metabolites of arachidonic acid, HODE– of linoleic acid, whereas 12-HEPE is a derivative of EPA. 12-HETE and HODE were the major eicosanoids detected in tested rat serum (Table 4, Figure 3). The highest level of 12-HETE (about 857 ng/mL) was noted in the group supplemented with linseed oil, followed by olive oil (765 ng/mL) and rice oil (723 ng/mL) supplementation. When it comes to HODE, the highest concentration of this isoform was observed in serum of the group supplemented with sunflower oil (467 ng/mL). It was significantly increased when comparing to the animals fed olive oil (302.9 ng/mL) or carotino oil (260 ng/mL).

5-HETE was the third hydroxy fatty acid in terms of the content (Table 4, Figure 3). Its highest concentration was detected in serum of the standard diet fed rats (262 ng/mL). This result was significantly higher, comparing to any of the groups supplemented with one of the oils. The contents of the last two hydroxy fatty acids (15-HETE and 12-HEPE) were much lower. They ranged from 19 ng/mL in the carotino oil supplemented group and 22 ng/mL in the group fed with sunflower oil to 37 ng/mL (12-HEPE).

We also checked whether there were correlations between serum PUFA content and their metabolite concentrations in the groups studied (Table 5). We found interesting, strong correlations between the studied compounds. In the linseed oil-fed group 12-HEPE, belonging to EPA (C20:5, *n*-3) metabolites, negatively correlated with both linoleic and arachidonic acid contents. However, these eicosanoids correlated positively with its precursor—EPA. What is more, the negative correlation was also observed between arachidonic acid metabolites—15-HETE and 5-HETE and linoleic acid in this group (Table 5). Moreover, both linoleic and arachidonic acid derivatives—HODE and 15-HETE, respectively—were negatively correlated with these PUFA in the rice oil-fed group. On the contrary, we observed strong positive correlation between arachidonic acid and its two metabolites—15-HETE and 5-HETE—in the sunflower oil supplemented group. The concentration of the third of tested arachidonic acid metabolites, 12-HETE, was also positively associated with arachidonic acid content, but this result was not significant. Yet, it was still on a trend level with *p* value = 0.053 (Table 5).

### 3.4. Growth of Rats Fed Various Oils

The growth rate of rats supplemented with studied oils was depicted in the Figure 4. It was observed the supplementation slowed down the rat growth through 4 weeks of experiment, compared to the control group. The smallest increase of body weight was observed in the olive oil and linseed oil supplemented groups, whereas the highest was observed in the sesame oil fed group followed by the group receiving the rice oil (Table 6, Figure 5).

The supplementation with oils caused only slight changes in weight of rat organs. No differences were noted in weights of livers of the animals fed edible oils (Table 6, Figure 6A), while statistically significant alterations, in the result of oil supplementation, were found in the weight of spleens (Table 6, Figure 6B) and kidneys (Table 6, Figure 6C). When it comes to spleens, the lowest mean values of their weight were noted in the animals fed the diet supplemented with linseed oil, olive oil, rice oil, and sunflower oil (0.4 g each) (Table 6, Figure 6B). They were significantly lower, when compared to the groups supplemented with carotino or sesame oils, as well as to the control group. Similarly, statistically lower mean value of kidneys was also observed in the group supplemented with olive oil (1.4 g), comparing to the control group. Kidneys of olive oil supplemented rats generally had the lowest weight, however it was not significant in comparison with the other groups (Table 6, Figure 6C).

## 4. Discussion

The level of fatty acids in the body can be affected by many factors such as a diet (including intake of vegetable oils, characterized by a diverse fatty acid composition), the activity of PUFA transforming enzymes (desaturases and elongases), and overall health. The impact of various edible oils on fatty acid profile and metabolites of arachidonic, linoleic, and eicosapentaenoic acids in rat serum, as well as on activity of Δ6- and Δ5-desaturases was studied. This is not the first study on the effect of oils on fatty acids and their derivatives [34,37]. For the first time, however, it is so comprehensive and this is where its uniqueness lies. So far, the influence of popular, commonly used oils—such as sunflower, olive, or soybean oils—on the content of fatty acids and their derivatives has been studied. In our previous works we investigated the relation between conjugated linoleic acid (CLA) and fatty acid profile and metabolites [38]. This time we focused on vegetable oils not so popular in Western countries as the above-mentioned ones, but of still increasing consumer interest. We chose cartino, rice, and sesame oils as such oils to compare their impact with well-known fats, like sunflower, olive, and linseed oils. What is more, prostaglandin E_2_ (PGE_2_), synthesized from arachidonic acid on the cyclooxygenase pathway, is the most often determined and well investigated metabolite of polyunsaturated fatty acids. On the contrary, in the current study, we analyzed the impact of the supplemented oils on PUFA metabolites synthesized on LOX pathway. They play an equally important role in the human body as COX derivatives. This is another aspect of uniqueness of this research.

The vegetable oils used in the study differ in the content of individual fatty acids as well as in the proportions of *n*-6 and *n*-3 PUFA. Some authors pointed out that this relationship may have a direct impact on the fatty acid profile in the blood serum [33,34,39].

With regard to the saturated fatty acids, palmitic and stearic acids are the most common representatives of this group in the human body. As they are synthesized in humans, their dietary intake, especially of palmitic acid, may not always significantly influence serum contents [40]. In our results, there were no significant differences in the percentage of these acids of all oil-fed groups compared to the control one. However, statistical disparities were found among the oil-supplemented groups. The most interesting observation concerns the group receiving rice oil. Serum palmitic acid concentration was the lowest (13.6%), whereas stearic acid was the highest (about 21.3%) in this group, although the rice oil contains nearly 20% of palmitic acid. This relation can be easily explained by palmitic acid metabolism. Under normal physiological conditions, C16:0 is elongated to stearic acid or desaturated to palmitooleic acid (C16:1), what also explicates elevated stearic acid level in the rice-oil-fed group [40]. Nevertheless, the ratios among saturated, monounsaturated, and polyunsaturated fatty acids seem to remain unchanged regardless the supplementation used. Probably the profile of individual fatty acid families is quite stable and does not change after the introduction of dietary supplementation. The maintaining of the balance between saturated and unsaturated fatty acids is thought to be crucial for various physiological functions. The disruption of fatty acid ratios may lead to pathological states such as cardiovascular and neurodegenerative diseases and cancer [40]. Imbalanced saturated and unsaturated proportions are considered to promote various mechanisms resulting in elevated LDL cholesterol level and suppression of LDL receptor expression [41]. This results in the reduced removal of LDL from plasma. What is more, saturated fatty acids were found to stimulate expression of peroxisome-proliferator-activated receptor-γ coactivator 1β (PGC-1β) in liver, which activates expression of transcription factors related to the lipid synthesis in liver. Consequently, the synthesis, production and secretion of very-low-density lipoproteins (VLDL), rich in cholesterol and triglycerides, increase [42]. Saturated fatty acids may also influence the inflammatory response activating nuclear factor κB (NF-κB). This leads to the increased synthesis of proinflammatory cytokines such as interleukin 6 (IL-6) or tumor necrosis factor α (TNF-α) as well as to overexpression of cyclooxygenase-2 (COX-2), involved in the biosysthesis of proinflammatory eicosanoids from arachidonic acid [43,44]. In our experiment, the rice oil given to rats contributed to the greatest changes in palmitic and stearic acid contents, despite not affecting the overall saturated fatty acid level. In the light of above information, the carotino oil seems to be an interesting proposition for consumers, because it decreases the total saturated fatty acid content and in this way may exert a protective activity.

Of the all fatty acids determined in rat serum, MUFA were found in the smallest proportion with oleic acid being the main representative of this family. The highest contents of oleic acid were noted in the carotino and olive oil-supplemented groups (12.7% and 9.9%, respectively). They were significantly higher comparing to the groups fed the rice, sesame, and linseed oils (Table 3). Olive oil and carotino oil are the main sources of oleic acid (73–75% and 53%, respectively), leaving the remaining oils used in the study behind (Table 2) [45]. What is more, olive oil is the major fat of the Mediterranean countries and, together with the whole lifestyle of this area, has been the subject of many studies [46,47]. Oleic acid does not belong to essential fatty acids, because it is synthesized in humans by stearoyl-CoA desaturase 1 (SCD1) from stearic acid. Nevertheless, its health role appears to be very interesting and has been investigated by many authors [48]. The beneficial impact of oleic acid was observed in cardiovascular diseases or rheumatoid arthritis. The consumption of this fatty acid increased the content of anti-inflammatory leukotriene A3 (LTA3), which is a potent inhibitor of pro-inflammatory leukotriene B4 (LTB4) and in this way prevent rheumatoid arthritis development [49]. Oleic acid was also found to reduce organ dysfunction and mortality of experimental sepsis in mice by inducing fatty acid oxidation, decreasing plasma non-estrified fatty acid concentration, the reactive oxygen species synthesis, and production of pro-inflammatory cytokines [50,51]. Treatment with oleic acid also inhibited neutrophile migration and accumulation in infected sites in mice [51]. The moderate consumption of olive oil seems to have the beneficial effect on the risk of breast cancer [46].

As regards PUFA, arachidonic acid followed by linoleic acid, both belonging to the *n*-6 family, dominated in the all groups. In humans, linoleic acid taken from the diet is metabolized first by D6D to GLA, which is elongated to DGLA. DGLA is then converted by D5D to arachidonic acid. The same enzymes convert ALA to EPA, which is further metabolized by D6D, elongases, and β-oxidated to DHA. Therefore, we expected the highest contents of arachidonic and linoleic acids in the groups supplemented with oils richest in linoleic acid—sunflower oil, followed by sesame and rice oils, i.e., the oils we have been particularly focused on. However, not entirely in line with our expectations and our previous studies [33,34], the highest arachidonic acid level was found in the rice oil-fed group. This value was significantly increased than in the carotino oil-supplemented group and was followed by the group fed with sesame oil. The arachidonic acid contents we have determined are higher compared to some other studies [52]. However, results similar to ours were also described by other authors [53]. This may be due to the increased endogenous synthesis of this fatty acid in rats, particularly under conditions of increased linoleic acid supply. The activity of enzymes involved in linoleic acid metabolic pathway can also influence arachidonic acid synthesis.

Regarding serum linoleic acid, its highest content was observed in the control group, whereas it was slightly lower in all the oil-supplemented groups. The lowest LA concentration was found in serum of rats receiving olive oil (11.3%), what reflects its composition. Olive oil contains the lowest amount of linoleic acid (6%) of all the oils tested [54]. As far as *n*-3 PUFA are concerned, DHA was present in the highest contents. However, they do not exceed 3.5%. DHA precursors—ALA and EPA—were detected in small amounts, below 1% even in the group fed with linseed oil, which is the best source of ALA of all oils tested [55].

The results concerning PUFA may be explained by both desaturase activities and synthesis of eicosanoids. Supplementation with each of the oils significantly intensified D6D activity, comparing to the control group, which resulted in increased GLA contents in the oil-fed groups. Most of the oils also increased the D5D activity, which was, to our surprise, the highest in the olive oil-receiving group, followed by the groups supplemented with rice, linseed and sesame oils.

The serum fatty acid profile depends on their dietary composition. However, the results obtained in this study indicate that the other factors, such as endogenous synthesis or the activity of enzymes involved in PUFA metabolism, may also affect the fatty acid profile in the body. In the present study, the concentrations of five eicosanoids—5-, 12-, 15-HETE, HODE, and 12-HEPE—in serum of rats fed with one of vegetable oils were determined (Table 4). 12-HETE was the major of detected compounds, regardless of the supplementation used. Its highest concentration was observed in serum of rats supplemented with linseed oil (857 ng/mL, Table 4, Figure 3). This result is rather unexpected because linseed oil is a very good source of α-linolenic acid and contains only approx. 15% of linoleic acid, which is the precursor of arachidonic acid. However, our finding may be explained by the not entirely obvious metabolism of linoleic and linolenic acids [56]. ALA conversion to its metabolites EPA and DHA may be affected by various factors, such as gender or diet. It is well known that diet rich in linoleic acid decreases conversion of α-linolenic acid by even 40% [57]. The conversion rate may be also decreased by high intake of EPA and DHA and by high intake of α-linolenic acid itself [58]. This fact appears to refer to our results and probably helps to explain them. The high level of 12-HETE in serum of linseed oil fed rats corresponds to a relatively high concentration of linoleic and arachidonic acid (Table 3, Figure 1), as well as to increased D6D and D5D activities (Figure 2). The similar situation concerns results from olive oil supplemented rats, where 12-HETE level was high (765 ng/mL serum), despite the low proportion of linoleic acid in olive oil (approx. 6%) (Table 2).

Sunflower, sesame, and rice oils contain the highest contents of linoleic acid—approximately 60%, 44%, and 30%, from among tested oils respectively (Table 2). In this case, we should have expected to find the higher levels of arachidonic acid metabolites in serum of rats supplemented with one of these oils. Meanwhile, this poor influence of dietary fats on the content of the main hydroxy fatty acid in serum may result from relatively short supplementation time (35 days) and considerable range of results within one group. Similar situation has been described by other authors who determined the content of eicosanoids in blood of salmon fed with diets containing either sunflower oil or fish oil [59]. Sunflower oil administration, despite high content of linoleic acid, reduced not only the levels of EPA metabolites—12-HEPE and LTB5—but also arachidonic acid metabolite—12-HETE [59]. The synthesis of eicosanoids derived from arachidonic acid can be inhibited by DGLA [60]. The decreased content of arachidonic acid derivatives in sunflower oil-supplemented group may also be due to 12-HETE oxidation to 12-oxo-eicosatetraenoic acid (12-oxo-HETE) [61].

Linoleic acid derivatives—HODE—were present in the samples in slightly smaller amounts than 12-HETE. The highest concentrations was found in the group of animals supplemented with sunflower oil (468 ng/mL, Table 4, Figure 3). It was significantly more than in serum of rats receiving olive oil (303 ng/mL) or carotino oil (260 ng/mL). Sunflower oil is a good source of linoleic acid (approximately 60%), which may be converted by 15-LOX-1 to HODE [62]. For comparison, the content of linoleic acid in olive oil and carotino oil was 5.9% and 16.8%, respectively. HODE concentration in the group supplemented with carotino oil was also significantly lower in comparison to animals fed only with feed (401 ng/mL, Table 4, Figure 3). We can therefore conclude that serum HODE concentrations in the best way reflects linoleic acid content in the diet, comparing to other eicosanoids. A similar conclusion was produced by Ferdouse et al. who observed that higher dietary linoleic acid increased its hydroxy metabolites, but did not influence arachidonic acid derivatives [63].

12-HEPE, which is an EPA derivative, was determined in the tested samples. It was the only detectable compound among HEPE isomers. Its concentrations appear to be probably modulated by dietary γ-linolenic acid, which corresponds to results received by other researchers [63]. Ferdouse et al. observed that HEPE hydroxy derivatives can be increased not only by dietary EPA, but also by α-linolenic acid and DHA [63].

What is more, we found strong correlations between serum PUFA content and their metabolite concentrations (Table 5). However, other studies have shown that that there are not always correlations between PUFA and their hydroxyl metabolites [64]. In our work 15-HETE, 5-HETE, and 12-HETE concentrations negatively correlated with linoleic acid, what results from their various metabolic pathways. 12-HEPE was also negatively correlated with arachidonic acid content, while strong positive correlation was observed between 12-HEPE and its precursor—EPA. Furthermore, we observed strong positive correlation between arachidonic acid and its three metabolites—15-HETE, 5-HETE, and 12-HETE (the last result despite of being not significant, it was still on a trend level). These correlations confirm different and complicated pathways of PUFA metabolism and indicate these processes are subjected to the strict endogenous control.

Fatty acids are thought to play a crucial role in people health. The disturbances of their profile and metabolites may be the basis of various disorders, like inflammation and its results—heart failure, asthma, a neoplastic process. What is more, the knowledge of these relations may help to predict the risk of one of above-mentioned diseases. Fatty acids derivatives appear to be influenced by nutrition. An appropriate composition of the diet affects directly or indirectly—through regulation of gene suppression—synthesis of fatty acid metabolites, such as eicosanoids. They may reflect the current state of an organism and constitute an interesting tool for foreseeing and inflammation-based disorders [65]. Both experimental and epidemiological investigations suggest beneficial impact of *n*-6 and *n*-3 PUFA in arythmia, hypertension, and atherothrombotic cardiovascular disease. The results of a randomized, double-blind, placebo-controlled study performed on nearly 7000 participants confirmed that supplementation with 1 g of *n*-3 PUFA daily significantly reduced cardiovascular mortality and admission to hospital for cardiovascular reasons in patients with heart failure [66]. This and other studies have been recently summarized by Sakamoto et al. who highlighted some mechanisms of *n*-3 PUFA protective role in heart failure [67]. They involve direct and indirect actions, such as alterations of metabolic cardiomyocyte conditions or anti-inflammatory properties of PUFA metabolites. *N*-3 PUFA also appeared to be protective in various neurodegenerative diseases, like Parkinson and Alzheimer’s diseases or depression, however the results are unequivocal [68,69]. Nevertheless, prospective studies carried out in Europe and USA showed decreased risk of Alzheimer’s disease and dementia due to increased *n*-3 PUFA intake [14,70]. Consumption of fatty fish more often than twice a week also correlated with the reduced risk of Alzheimer’s disease (by 41%) [71]. Carcinogenesis is another process in which fatty acids, especially unsaturated ones, play an important role. Lipids were observed to accumulate in many cancer tissues, including colorectal, breast, brain, and ovarian [72]. It has been recently found that lipid droplets are sites where enzymes converting fatty acids to their metabolites—eicosanoids—are located and where the synthesis of eicosanoids takes place during inflammation and cancer [73]. These discoveries make both enzymes involved in the metabolism of fatty acids and metabolites interesting factors that can modify the course of neoplastic activity.

## 5. Conclusions

In conclusion, the results received indicated that edible oils supplemented to rats daily influence serum fatty acid profile and concentrations of metabolites of arachidonic and linoleic acid and EPA. They also affect activity of enzymes involved in metabolism of fatty acids. However, not always both fatty acid profile and HETE, HODE, and HEPE levels reflect fatty acid composition of supplemented oils. Based on the results obtained, it is also very difficult to indicate which of the oils can have the most beneficial health effect. Of the oils we were particularly interested in this study, rice and sesame oils had the highest *n*-6 PUFA contents in serum. The role of this PUFA family in various pathological processes was mentioned above. On the contrary, the lowest *n*-6 PUFA levels were observed in olive and carotino oils which also had the highest content of *n*-3 PUFA. All these facts show the complexity of lipid metabolism that leads to synthesis of compounds playing various role in the development or prevention of many pathological processes. The profile of serum fatty acids and their metabolites may reflect the present state of an organism and constitute an interesting tool for prevention and therapy of many inflammation-based disorders.

## Figures and Tables

**Figure 1 nutrients-12-01232-f001:**
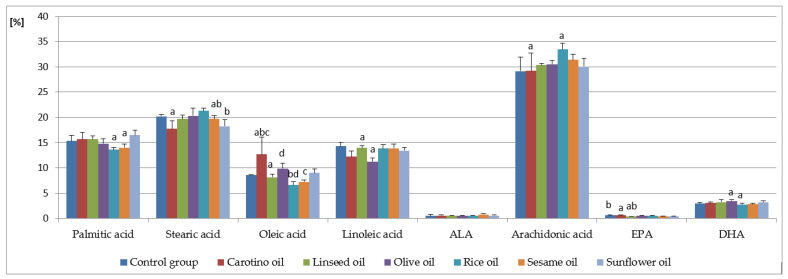
Contents of selected fatty acids in serum of rats supplemented with selected oils (%). Data were expressed as mean values ± standard deviation. Bars with the same superscript (a, b, c, or d) above referring to one fatty acid are significantly different in one-way ANOVA followed by RIR Tukey’s post-hoc test (*p* < 0.05). ALA—α-linolenic acid; EPA—eicosapentaenoic acid; DHA—docosahexaenoic acid.

**Figure 2 nutrients-12-01232-f002:**
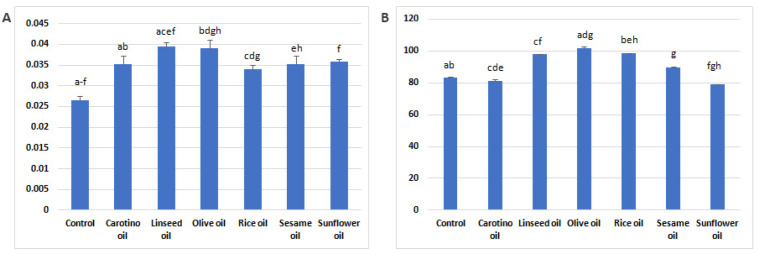
Activities of D6D (**A**) and D5D (**B**) in rat serum. All data are shown as mean values ± standard deviation. Bars with the same superscript (a–h) above are significantly different in one-way ANOVA followed by RIR Tukey’s post-hoc test (D6D) or Kruskal–Wallis test followed by multiple comparison test (D5D) (*p* ≤ 0.05). D6D—Δ6-desaturase index, D5D—Δ5-desaturase index.

**Figure 3 nutrients-12-01232-f003:**
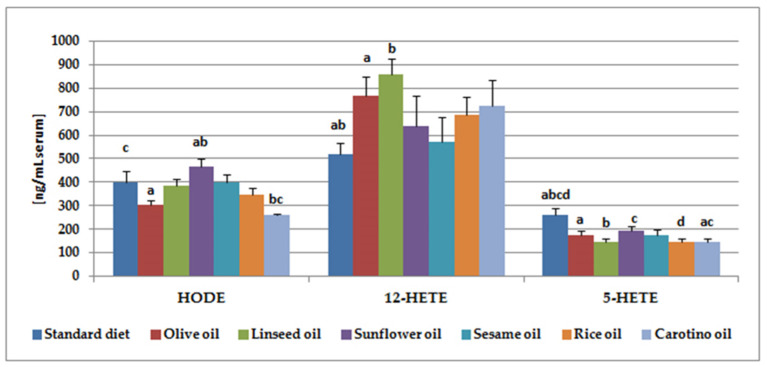
Contents of selected fatty acid metabolites in serum of rats supplemented with edible oils (ng/mL serum). Data were expressed as mean values ± standard error of mean. Bars with the same superscript (a, b, c, or d) above referring to one metabolite are significantly different (*p* < 0.05).

**Figure 4 nutrients-12-01232-f004:**
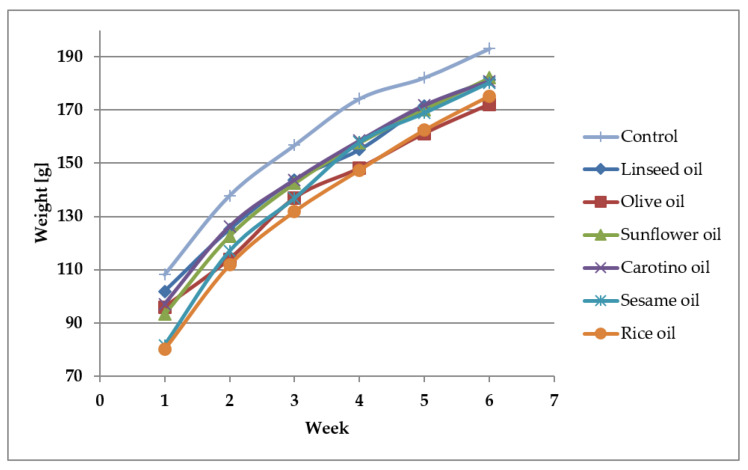
Growth rate of rats supplemented with selected edible oils.

**Figure 5 nutrients-12-01232-f005:**
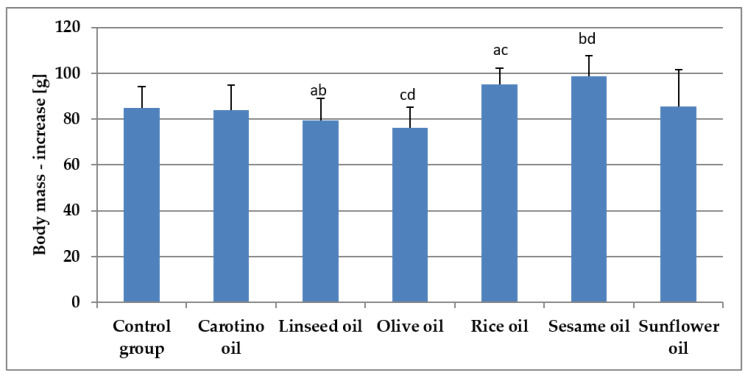
Body mass increase (g) of rats supplemented with selected oils. Bars with the same superscript (a, b, c, or d) above are significantly different (*p* < 0.05).

**Figure 6 nutrients-12-01232-f006:**
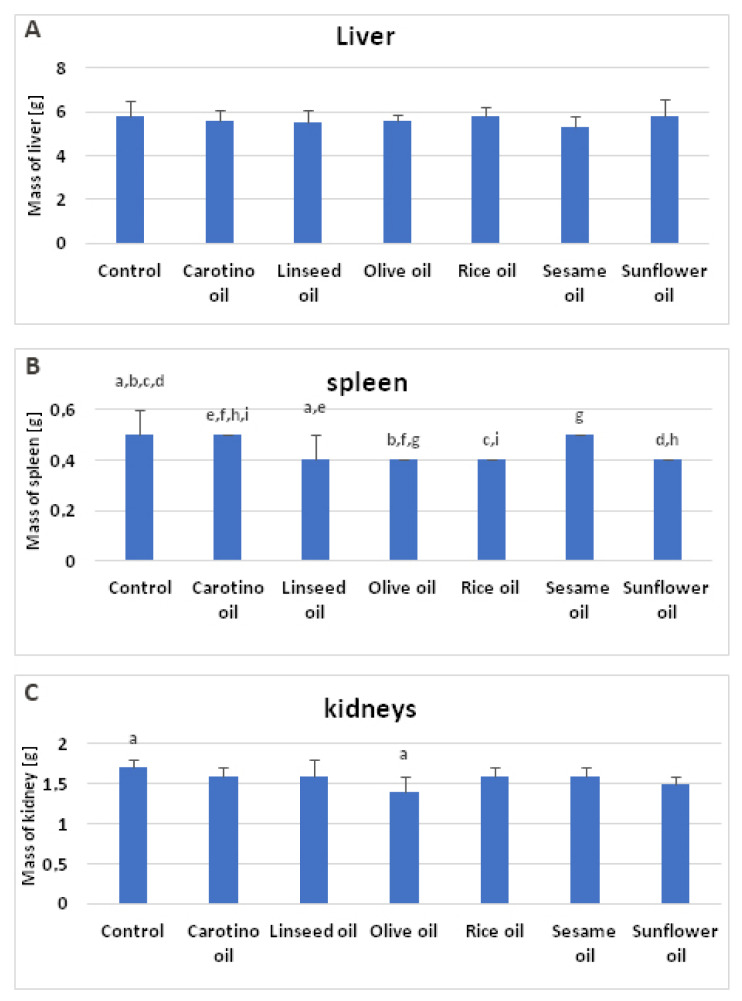
Organ weights of rats supplemented with selected oils (**A**—mass of liver; **B**—mass of spleen; **C**—kidneys). Bars with the same superscript (a–i) above are significantly different (*p* < 0.05).

**Table 1 nutrients-12-01232-t001:** Declared dietary composition of Labofeed H (per kg of diet)

Protein (g)	210.0		
Fat (g)	39.2		
Fiber (g)	43.2		
Starch (g)	300.0		
Ash (g)	55.0		
Vitamin A (IU)	15,000	Vitamin B_6_ (mg)	17.0
Lysine (g)	14.5	Histidine (g)	6.0
Vitamin D_3_ (IU)	1000	Vitamin B_12_ (µg)	80.0
Methionine (g)	4.1	Arginine (g)	13.0
Vitamin E (mg)	90.0	Pantothenate (mg)	30.0
Tryptophan (g)	3.0	Phenylalanine (g)	10.0
Vitamin K_3_ (mg)	3.0	Folic acid (mg)	5.0
Threonine (g)	7.4	Tyrosine (g)	7.8
Vitamin B_1_ (mg)	21.0	Nicotinic acid (mg)	133.0
Isoleucine (g)	17.5	Choline (mg)	2750.0
Vitamin B_2_ (mg)	16.0	Biotin (mg)	0.4
Valine (g)	11.0		
Calcium (g)	10.0	Potassium (g)	9.4
Iron (mg)	250.0	Cobalt (mg)	2.0
Phosphorus total (g)	8.17	Sodium (g)	2.2
Manganese (mg)	100.0	Iodine (mg)	1.0
Phosphorus saturated (g)	4.5	Chlorine (g)	2.5
Zinc (mg)	76.9	Selenium (mg)	0.5
Magnesium (g)	3.0	Sulfur (g)	1.9
Copper (mg)	21.3		

**Table 2 nutrients-12-01232-t002:** Fatty acid composition of oils used in the experiment (%).

Fatty Acid	Olive Oil	Linseed Oil	Sunflower Oil	Sesame Oil	Carotino Oil	Rice Oil
Myristic acid (C14:0)	0.02	nd	0.06	0.02	0.35	0.37
Palmitic acid (C16:0)	10.92	5.44	5.97	10.00	13.59	19.88
Stearic acid (C18:0)	2.58	2.86	4.91	5.21	2.50	2.07
Oleic acid (C18:1 n-9)	74.46	11.29	25.40	36.27	52.94	40.90
Linoleic acid (C18:2 n-6)	5.94	15.47	59.91	44.21	16.85	29.75
α-Linoleic acid (C18:3 n-3)	0.61	63.98	0.11	0,35	0.57	0.11
Eicosanoic acid (C20:0)	0.45	0.15	0.28	0.54	0.49	0.81

Nd—not detected.

**Table 3 nutrients-12-01232-t003:** Fatty acid composition in serum of rats supplemented with selected oils (%)

Fatty Acids (%)	C	CAR	L	O	R	SES	SUN
**SFA**							
Lauric acid (C12:0)	0.07 ± 0.03	0.05 ± 0.01	0.10 ± 0.05	0.06 ± 0.03	0.02 ± 0.00	0.06 ± 0.00	0.11 ± 0.10
Myristic acid (C14:0)	0.37 ± 0.03	0.39 ± 0.04	0.40 ± 0.02	0.44 ± 0.16	0.40 ± 0.06	0.41 ± 0.00	0.45 ± 0.09
Pentadecanoic acid (C15:0)	0.31 ± 0.04	0.34 ± 0.02	0.32 ± 0.03	0.29 ± 0.02	0.34 ± 0.05	0.35 ± 0.02	0.36 ± 0.03
Palmitic acid (C16:0)	15.31 ± 1.10	15.66 ± 1.31	15.65 ± 0.69	14.78 ± 0.94	13.61 ± 0.40 ^a^	13.98 ± 0.69	16.47 ± 0.98 ^a^
Heptadecanoic acid (C17:0)	0.63 ± 0.03 ^a^	0.45 ± 0.05 ^ab^	0.56 ± 0.03	0.55 ± 0.06	0.57 ± 0.03 ^b^	0.50 ± 0.06	0.55 ± 0.07
Stearic acid (C18:0)	20.20 ± 0.32	17.72 ± 1.60 ^a^	19.70 ± 0.77	20.28 ± 1.54	21.27 ± 0.60 ^ab^	19.65 ± 0.72	18.25 ± 1.25 ^b^
Arachidic acid (C20:0)	0.05 ± 0.00	0.05 ± 0.02	0.06 ± 0.01	0.05 ± 0.01	0.06 ± 0.01	0.07 ± 0.01	0.06 ± 0.01
Lignoceric acid (C24:0)	0.13 ± 0.01 ^a^	0.15 ± 0.02	0.16 ± 0.02	0.15 ± 0.02	0.14 ± 0.02	0.14 ± 0.01	0.18 ± 0.02 ^a^
Σ SFA	37.06 ± 1.18	34.85 ± 0.97 ^a^	36.93 ± 0.78 ^a^	36.60 ± 0.57	36.46 ± 0.85	35.52 ± 1.01	36.50 ± 0.89
**MUFA**							
Myristoleic acid (C14:1)	0.04 ± 0.01	0.04 ± 0.01	0.02 ± 0.01	0.02 ± 0.01	0.04 ± 0.02	0.05 ± 0.01	0.03 ± 0.00
Palmitoleic acid (C16:1)	1.03 ± 0.01	1.53 ± 0.75 ^a^	1.04 ± 0.11	1.12 ± 0.15	0.78 ± 0.09 ^ab^	1.55 ± 0.36 ^b^	1.18 ± 0.27
Heptadecenoic acid (C17:1)	0.10 ± 0.02	0.13 ± 0.03	1.12 ± 0.05	0.11 ± 0.01	0.08 ± 0.01 ^a^	0.10 ± 0.02	0.15 ± 0.02 ^a^
Oleic acid (C18:1 *n*-9)	8.56 ± 0.07	12.70 ± 3.35 ^abc^	8.15 ± 0.55 ^a^	9.89 ± 1.03 ^d^	6.60 ± 0.60 ^bd^	7.21 ± 0.37 ^c^	9.00 ± 0.72
Eicosenoic acid (C20:1)	0.11 ± 0.02	0.12 ± 0.05	0.08 ± 0.03	0.13 ± 0.06	0.09 ± 0.01	0.12 ± 0.03	0.09 ± 0.02
Nervonic acid (C24:1 *n*-9)	0.41 ± 0.00 ^a^	0.33 ± 0.03	0.27 ± 0.03 ^ab^	0.31 ± 0.03	0.32 ± 0.03	0.30 ± 0.05	0.35 ± 0.03 ^b^
Σ MUFA	11.03 ± 1.37	13.42 ± 5.67 ^a^	9.67 ± 0.64	11.56 ± 1.18	7.92 ± 0.67 ^a^	9.42 ± 0.74	10.93 ± 0.98
**PUFA**							
Linolelaidic acid (C18:2 *n*-6 trans)	1.59 ± 0.22	1.67 ± 0.33	1.44 ± 0.07	1.51 ± 0.10	1.24 ± 0.36	1.55 ± 0.42	1.49 ± 0.14
Linoleic acid (C18:2 *n*-6)	14.30 ± 0.81	12.22 ± 1.09	13.94 ± 0.44 ^a^	11.26 ± 0.70 ^a^	13.80 ± 0.85	13.86 ± 0.83	13.40 ± 0.57
γ-Linolenic acid (C18:3 *n*-6)	0.38 ± 0.08 ^b^	0.43 ± 0.04 ^a^	0.55 ± 0.03 ^ab^	0.44 ± 0.03	0.47 ± 0.07	0.49 ± 0.07	0.48 ± 0.03
α-Linolenic acid (C18:3 *n*-3)	0.51 ± 0.37	0.57 ± 0.13	0.55 ± 0.09	0.49 ± 0.13	0.54 ± 0.07	0.77 ± 0.21	0.56 ± 0.10
Dihomo-γ-linolenic acid (DGLA, C20:3 *n*-6)	0.35 ± 0.04	0.36 ± 0.05	0.31 ± 0.07	0.30 ± 0.00	0.34 ± 0.05	0.35 ± 0.12	0.38 ± 0.06
Arachidonic acid (C20:4 *n*-6)	29.12 ± 2.79	29.21 ± 3.51 ^a^	30.32 ± 0.37	30.50 ± 0.75	33.42 ± 1.25 ^a^	31.40 ± 1.06	30.05 ± 1.62
Eicosapentaenoic acid (EPA, C20:5 *n*-3)	0.64 ± 0.03 ^b^	0.63 ± 0.09 ^a^	0.37 ± 0.04 ^ab^	0.55 ± 0.09	0.49 ± 0.06	0.39 ± 0.09	0.48 ± 0.05
Docosahexaenoic acid (DHA, C22:6 *n*-3)	2.93 ± 0.18	3.03 ± 0.24	3.20 ± 0.48	3.44 ± 0.26 ^a^	2.74 ± 0.29 ^a^	2.82 ± 0.22	3.21 ± 0.27
Σ PUFA	49.81 ± 1.69	49.41 ± 1.27 ^a^	50.77 ± 1.36	48.23 ± 1.45 ^b^	53.27 ± 1.11 ^ab^	51.45 ± 2.56	50.06 ± 1.26
*n*-3	4.08 ± 0.25	4.38 ± 0.09	3.84 ± 0.80	4.47 ± 0.30	3.77 ± 0.34	3.98 ± 0.24	4.26 ± 0.23
*n*-6	45.73 ± 1.88	45.07 ± 1.29 ^a^	46.94 ± 0.88	43.76 ± 1.29 ^b^	49.82 ± 0.53 ^ab^	47.47 ± 2.32	45.80 ± 1.13

Data were expressed as mean values ± standard deviation. Values in a row with the same superscript (a, b, c or d) are significantly different in one-way ANOVA followed by RIR Tukey’s post-hoc test (*p* < 0.05). C—control group; CAR—carotino oil; L—linseed oil; O—olive oil; R —rice oil; SES—sesame oil; SUN—sunflower oil.

**Table 4 nutrients-12-01232-t004:** Contents of arachidonic, linoleic, and eicosapentaenoic acid metabolites in serum of rats supplemented with diverse edible oils (ng/mL serum)

Hydroxyfatty Acid	Standard Diet	Olive Oil	Linseed Oil	Sunflower Oil	Sesame Oil	Rice Oil	Carotino Oil
**12-HEPE**	35.5 ± 4.8 ^ab^	37.5 ± 4.7	30.4 ± 3.2	22.1 ± 3.8 ^a^	22.9 ± 3.4 ^b^	26.4 ± 2.4	34.7 ± 5.9
**HODE**	400.9 ± 46.0 ^c^	302.9 ± 21.6 ^a^	383.3 ± 31.5	467.8 ± 31.3 ^ab^	398.3 ± 36.4	348.5 ± 29.2	259.9 ± 6.9 ^bc^
**15-HETE**	32.0 ± 6.1	26.3 ± 3.8	28.6 ± 2.1	39.+6 ± 4.0 ^a^	46.8 ± 8.0 ^b^	31.0 ± 3.5	19.1 ± 2.6 ^ab^
**12-HETE**	516.7 ± 51.2 ^ab^	765.2 ± 85.4 ^a^	857.3 ± 66.2 ^b^	638.0 ± 127.7	571.6 ± 104.2	683.9 ± 77.3	723.0 ± 110.7
**5-HETE**	262.4 ± 28.6 ^abcd^	175.8 ± 17.5 ^a^	147.2 ± 13.2 ^b^	195.2 ± 17.1^c^	176.3 ± 19.8	145.0 ± 13.0^d^	145.0 ± 13.0 ^ac^

Data were expressed as mean values ± standard error of mean. Values in a row with the same superscript (a, b, c or d) are significantly different in Kruskal–Wallis test followed by multiple comparison test (*p* < 0.05).

**Table 5 nutrients-12-01232-t005:** Correlation coefficients (r) between linoleic and arachidonic acid contents and their metabolites in rat serum

Correlation	Spearman’s Correlation Coefficient (r)	*p*-Value
**Linseed oil**		
12-HEPE vs. Linoleic acid	−0.980	0.0033
12-HEPE vs. Arachidonic acid	−0.850	0.0684 ^*)^
12-HEPE vs. EPA	0.882	0.0480
15-HETE vs. Linoleic acid	−0.881	0.0481
5-HETE vs. Linoleic acid	−0.930	0.0221
**Sunflower oil**		
15-HETE vs. Arachidonic acid	0.817	0.0249
12-HETE vs. Arachidonic acid	0.748	0.0532^*)^
5-HETE vs. Arachidonic acid	0.905	0.0051
**Rice oil**		
HODE vs. Linoleic acid	−0.966	0.0074
HODE vs. Arachidonic acid	−0.945	0.0151
15-HETE vs. Linoleic acid	−0.993	0.0006
15-HETE vs. Arachidonic acid	−0.888	0.0439

Only statistically significant results (*p* < 0.05) are given in the table with two exceptions where the results were on a trend level (^*)^). 12-HEPE—12-hydroxyeicosapentaenoic acid; HETE—hydroxyeicosatetraenoic acid; HODE—hydroxyoctadecadienoic acid.

**Table 6 nutrients-12-01232-t006:** Body and organ weights of rats supplemented with selected oils.

	Control Group	Carotino Oil	Linseed Oil	Olive Oil	Rice Oil	Sesame Oil	Sunflower Oil
Mass–start (g)	108.2 ± 11.0 ^ab^	97.0 ± 10.8	101.9 ± 16.9	95.9 ± 20.0	80.1 ± 12.9 ^a^	81.7 ± 10.7 ^b^	93.2 ± 7.5
Mass–end (g)	193.0 ± 14.8	180.9 ± 10.1	181.2 ± 19.5	172.1 ± 16.1	175.2 ± 13.1	180.3 ± 17.5	182.1 ± 20.4
Mass–increase (g)	84.8 ± 9.5	83.9 ± 10.8	79.3 ± 9.6 ^ab^	76.2 ± 8.9 ^cd^	95.1 ± 7.1 ^ac^	98.7 ± 8.9 ^bd^	85.5 ± 16.2
Liver (g)	5.8 ± 0.7	5.6 ± 0.5	5.5 ± 0.6	5.6 ± 0.3	5.8 ± 0.4	5.3 ± 0.5	5.8 ± 0.8
Liver (%)	3.0 ± 0.2	3.1 ± 0.1	3.0 ± 0.1	3.2 ± 0.3	3.3 ± 0.2	2.9 ± 0.3	3.2 ± 0.2
Spleen (g)	0.5 ± 0.1 ^abcd^	0.5 ± 0.0 ^efhi^	0.4 ± 0.1 ^ae^	0.4 ± 0.0 ^bfg^	0.4 ± 0.0 ^ci^	0.5 ± 0.0 ^g^	0.4 ± 0.0 ^dh^
Spleen (%)	0.3 ± 0.0	0.3 ± 0.0 ^ab^	0.2 ± 0.0 ^a^	0.2 ± 0.0 ^b^	0.2 ± 0.0	0.3 ± 0.0	0.2 ± 0.0
Kidneys (g)	1.7 ± 0.1 ^a^	1.6 ± 0.1	1.6 ± 0.2	1.4 ± 0.2 ^a^	1.6 ± 0.1	1.6 ± 0.1	1.5 ± 0.1
Kidneys (%)	0.9 ± 0.0	0.9 ± 0.0	0.9 ± 0.1	0.8 ± 0.0	0.9 ± 0.0	0.9 ± 0.1	0.8 ± 0.0

Data are expressed as mean values ± standard deviation. Values in a row with the same superscript (a, b, c or d) are significantly different (*p* < 0.05).

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
