# Peer review of "Oils’ Impact on Comprehensive Fatty Acid Analysis and Their Metabolites in Rats"

_nutrients, 2020, doi:10.3390/nu12051232_

Round 1
Reviewer 1 Report
In the present study, Stawarska et al., demonstrated that serum fatty acid levels in rats with 7 different chows. In addition, they also determined the activity of desaturates, and limited number of PUFA metabolites. Although their approach is interesting, there are several concerns and limitations needed to be addressed.
Comments
1. Fatty acid composition
As far as I know, the most abundant fatty acids is not arachidonic acid in rat serum but palmitic acid or oleic acid. Authors should discuss why fatty acid composition in the present study was different from previous works.
2.term of lipidomics
Lipidomics usually used for global lipid analysis. However, authors determined only fatty acids and their limited derivatives. Authors should use another term such as comprehensive fatty acid analysis instead of lipidomics.
3. Prostaglandins
Prostaglandins, at least PGE2, can also be detectable in serum. It is prefer if authors put prostaglandin (at least PGE2) data.
4. correlation of PUFA and their metabolites
It is important and interesting whether serum levels of parental PUFA (such as arachidonic acid) and those of their metabolites (such as HETE). Author should examines their correlations.
5. statistical test for table 1
There should be p values for each comparison. Author should present all p values of the comparison.
Author Response
Responses to Reviewers – manuscript of Nutrients...
We are very grateful to have been given the opportunity to revise our manuscript of Nutrients-781274. We would like to thank both the Reviewers for their time and effort in reading and reviewing the article and for all valuable comments. We found them very useful in improving the paper. We made all the changes suggested by the Reviewers and marked them in yellow (file nutrients-781274_revised_with-marked_changes). We also included the second file without marked changes (nutrients-781274_revised). As a result of the revision, we corrected the description of the statistical methods. We hope you and the Reviewers will be satisfied with our revision.
Reviewer 1
In the present study, Stawarska et al., demonstrated that serum fatty acid levels in rats with 7 different chows. In addition, they also determined the activity of desaturates, and limited number of PUFA metabolites. Although their approach is interesting, there are several concerns and limitations needed to be addressed.
Response: Thank you very much for the time and effort you put into reviewing our manuscript. We are very grateful for all your valuable comments. We found them really helpful in improving our paper. Below we enclosed our responses to each of your items. We hope you will be satisfied with our revision.
Comments
- Fatty acid composition
As far as I know, the most abundant fatty acids is not arachidonic acid in rat serum but palmitic acid or oleic acid. Authors should discuss why fatty acid composition in the present study was different from previous works.
Response: Thank you for mentioning this point. We discussed this issue as you suggested.
- term of lipidomics
Lipidomics usually used for global lipid analysis. However, authors determined only fatty acids and their limited derivatives. Authors should use another term such as comprehensive fatty acid analysis instead of lipidomics.
Response: Thank you for your suggestion. We use the term “lipidomic” only in the title of the paper, trying to make it short but as comprehensive and communicative as possible at the time. However, since we really ojnly determined fatty acids and their selected metabolites, we have replaced the term “lipidomics” according to your hint.
- Prostaglandins
Prostaglandins, at least PGE2, can also be detectable in serum. It is prefer if authors put prostaglandin (at least PGE2) data.
Response: Thank you for mentioning this point. At the beginning of this study we unfortunately did not plan to determine PGE2, because we wanted to focus on hydroxy fatty acids, which are not so well known as PGE2. It later turned out that we did not have enough material for analyses. However, this parameter might provide an interesting complement to our results.
- correlation of PUFA and their metabolites
It is important and interesting whether serum levels of parental PUFA (such as arachidonic acid) and those of their metabolites (such as HETE). Author should examines their correlations.
Response: Thank you for your suggestion. According to it we carried out a statistical analysis of correlations between PUFA and their metabolites. We completed information about correlations in the Materials and Methods section, as well as in Results and Discussion.
- statistical test for table 1
There should be p values for each comparison. Author should present all p values of the comparison.
Response: The Table 1 shows the composition of feed fed to rats as presented by the manufacturer. You probably meant the other tables, like 3 or 4. In the first version of our manuscript we included a column called “p value” to this tables. It contained the results of F-test (F-statistics) used in Analysis of Variance (ANOVA), when its assumption were fulfilled or the results or Kruskal-Wallis test for nonparametric variables. These tests show that not all of the group means are equal (p value < 0.05 or less) but they do not indicate which groups differ from each other. P-values we presented in the table 3 and 4 were just the results of ANOVA or Kruskal-Wallis test.
If, however, the ANOVA F-test or Kruskal-Wallis test are significant, the analysis must be continued to determine which particular group means differ. For this purpose, the so-called post-hoc tests are performed, like Tukey’s test or multiple comparison test used in our study. P-values calculated in these tests indicate significant differences.
As you suggested, we tried to include p-values for all comparisons, but due to the large number of the groups and many substances tested we decided not to put all p-values. Instead, in the legend below the tables we indicate that p-values <0.05 are significant.

Reviewer 2 Report
The manuscript by authors Agnieszka Stawarska etc. studied the effect of edible oil supplements fatty acid profile and metabolites of arachidonic, linoleic and eicosapentaenoic acids, as well as on the activity of Δ6- and Δ5-desaturases. They reported significant changes these parameters. The experimental designs is reasonable. However, data presentation is obscure and hard to appreciate. A major revision will be required. I listed some of my comments below.
- Table 3, fatty acids name or abbreviations, e.g. stearic acid, palmitic acid should be provided to understand the result section. What is column p-value referred to?
- The letters for significant levels carried no information. More detail information should be given either in the legend or main text.
- Graph presentations should provided for table 3, 4,5,6.
- A conclusion in terms of which vegetable oil supplement produce better fatty acid profile should be suggested. Or a rank of the oils of health benefit should be analyzed.
Author Response
Responses to Reviewers
We are very grateful to have been given the opportunity to revise our manuscript of Nutrients-781274. We would like to thank both the Reviewers for their time and effort in reading and reviewing the article and for all valuable comments. We found them very useful in improving the paper. We made all the changes suggested by the Reviewers and marked them in yellow (file nutrients-781274_revised_with-marked_changes). As a result of the revision, we corrected the description of the statistical methods. We hope you and the Reviewers will be satisfied with our revision.
Reviewer 2
The manuscript by authors Agnieszka Stawarska etc. studied the effect of edible oil supplements fatty acid profile and metabolites of arachidonic, linoleic and eicosapentaenoic acids, as well as on the activity of Δ6- and Δ5-desaturases. They reported significant changes these parameters. The experimental designs is reasonable. However, data presentation is obscure and hard to appreciate. A major revision will be required. I listed some of my comments below.
Response: Thank you very much for the time and effort you put into reviewing our manuscript. We are very grateful for all your valuable comments. We found them really helpful in improving our paper. Below we enclosed our responses to each of your items. We hope you will be satisfied with our revision.
- Table 3, fatty acids name or abbreviations, e.g. stearic acid, palmitic acid should be provided to understand the result section. What is column p-value referred to?
Response: Thank you for mentioning this point. We completed Table 3 in full fatty acid names. Concerning the p-value column in this table, it contains the results of F-test (F-statistics) used in Analysis of Variance (ANOVA), when its assumption were fulfilled or the results or Kruskal-Wallis test for nonparametric variables. These tests show that not all of the group means are equal (p value < 0.05 or less) but they do not indicate which groups differ from each other.
If, however, the ANOVA F-test or Kruskal-Wallis test are significant, the analysis must be continued to determine which particular group means differ. For this purpose, the so-called post-hoc tests are performed, like Tukey’s test or multiple comparison test used in our study. P-values calculated in these tests indicate significant differences between particular groups. So to be precise we should have included the p-values obtained in the post-hoc tests. However, due to the large number of the groups and many substances tested we decided not to put all p-values at all in the table. Instead, in the legend below the table 3 we indicate that p-values <0.05 are significant.
- The letters for significant levels carried no information. More detail information should be given either in the legend or main text.
Response: We have improved the description of letters (superscripts) indicating significant differences in legends of each table. We assumed p-value < 0.05, which is the most often chosen value for statistically significance. Nevertheless, lower p-values were also calculated based on our results. If there is the same superscript (eg. “a”) besides the results from the two groups, these results differ significantly from each other. However, each superscript (letter) always indicates the probability value lower than 0.05.
- Graph presentations should provided for table 3, 4,5,6.
Response: Thank you for this advice. We provided graph presentations for the most important results from the tables 3-6. We think they complement the tables well.
- A conclusion in terms of which vegetable oil supplement produce better fatty acid profile should be suggested. Or a rank of the oils of health benefit should be analyzed.
Response: Thank you for this advice. We included discussion in the Conclusions section.

Round 2
Reviewer 1 Report
I think revised manuscript addressed all comments and became informative.